# A Ku-Band Broadband Stacked FET Power Amplifier Using 0.15 μm GaAs pHEMT

**DOI:** 10.3390/mi14061276

**Published:** 2023-06-20

**Authors:** Jiaxuan Li, Yang Yuan, Bin Yuan, Jingxin Fan, Jialong Zeng, Zhongjun Yu

**Affiliations:** 1Aerospace Information Research Institute, Chinese Academy of Sciences, Beijing 100094, China; lijiaxuan201@mails.ucas.ac.cn (J.L.); yuanyang19@mails.ucas.ac.cn (Y.Y.); yuanbin19@mails.ucas.ac.cn (B.Y.); fanjingxin21@mails.ucas.ac.cn (J.F.); zengjialong19@mails.ucas.ac.cn (J.Z.); 2School of Electronic, Electrical and Communication Engineering, University of Chinese Academy of Sciences, Beijing 101408, China

**Keywords:** gallium arsenide (GaAs), Ku-band, microwave monolithic integrated circuit (MMIC), broadband power amplifier, stacked FET

## Abstract

To meet the application requirements of broadband radar systems for broadband power amplifiers, a Ku-band broadband power amplifier (PA) microwave monolithic integrated circuit (MMIC) based on a 0.15 µm gallium arsenide (GaAs) high-electron-mobility transistor (HEMT) technology is proposed in this paper. In this design, the advantages of the stacked FET structure in the broadband PA design are illustrated by theoretical derivation. The proposed PA uses a two-stage amplifier structure and a two-way power synthesis structure to achieve high-power gain and high-power design, respectively. The fabricated power amplifier was tested under continuous wave conditions, and the test results showed a peak power of 30.8 dBm at 16 GHz. At 15 to 17.5 GHz, the output power was above 30 dBm with a PAE of more than 32%. The fractional bandwidth of the 3 dB output power was 30%. The chip area was 3.3 × 1.2 mm^2^ and included input and output test pads.

## 1. Introduction

The design challenge of broadband power amplifiers (PAs) lies in how to achieve a broadband design while ensuring good broadband output power characteristics. Wideband PAs can be adapted to various RF systems by covering multiple RF bands with a single chip. This in turn reduces the cost and complexity of multi-band and multi-mode systems by reducing the cost of the chip [1,2]. For this reason, multi-band or broadband PA design techniques have been an important area of research [3,4,5,6,7,8].

In [9,10], the design of a multi-band PA was achieved by using two different PAs to achieve different operating frequencies and, thus, multi-band PAs. However, the large size of the broadband PA designed in this way is not conducive to streamlining the size of the system, which in turn can raise the cost of the whole system. Distributed structures [11,12] are widely used for broadband PAs larger than two octaves. However, distributed power amplifiers (DPAs) have significant drawbacks in terms of power gain, which is typically only a few dB. In order to meet the engineering requirements (usually several tens of dB are required), additional broadband drive circuitry is required, which greatly increases the complexity and failure rate of the system. In order not to increase the complexity of the system, other broadband PA designs have been proposed in several studies. Wideband PAs with a 16.2%, 7.7%, and 18.7% fractional bandwidth by using multi-stage matching networks were reported in [13,14,15], respectively. However, since a multi-stage matching network requires more passive components than a single-stage matching network, the circuit size and the losses in the matching network will increase significantly.

In addition, the choice of semiconductor technology is also an important part of the process of designing power devices. There are several major semiconductor materials for power device design commonly used in the microwave-/millimeter-wave field, including gallium nitride (GaN), gallium arsenide (GaAs), and indium phosphide (InP). The HEMTs on GaAs substrates used in this design have several advantages [16]: (1) GaAs HEMTs have similar gain and noise performance and a lower fabrication cost compared to InP HEMTs. (2) Compared with GaN HEMTs, although GaAs HEMTs have lower power density, the process is more mature and has obvious advantages in terms of cost. However, GaAs HEMTs have limited application in high-power applications due to their lower breakdown voltage. Generally speaking, in order to increase the output power of PAs, the voltage swing or current swing of the amplifier needs to be increased. To increase the output power of the amplifier, the size of the device periphery is usually increased to allow for a larger current swing. The increase in device size will increase its maximum power, but will reduce its optimum load impedance, which will be difficult to match in a system with a 50-ohm load.

In order to solve the two problems mentioned above, which are detrimental to the design of broadband PAs, one advantageous approach lies in using stacked FET structures to increase the voltage swing and the output impedance. The use of stacked FET structures offers significant advantages in terms of operating bandwidth, power gain, and circuit size compared to the traditional parallel approach [17,18]. First of all, the load impedance required for optimal power matching is much larger in stacked FET structures (typically several times the load impedance of a single FET), which allows output matching to be achieved in a low-Q region [19,20]. In addition, stacked FET structures are one of the promising solutions to the low breakdown voltage problem. Due to the voltage summation effect, the voltage swing on the top transistor will be allowed to increase exponentially, yet the voltage swing on a single FET can still remain relatively small, thus alleviating the low-breakdown-voltage problem of GaAs HEMTs.

The broadband PA presented in this paper was implemented by a double-stacked transistor structure. The selection of the interconnection line length between the stages of the stacked transistors was analyzed based on the small-signal model. Meanwhile, the influence of the interconnection line on the output impedance matching is discussed theoretically. The final measured results showed a peak power of 30.8 dBm at 16 GHz and a peak power-added efficiency (PAE) of 41%. At 15 to 17.5 GHz, the output power was above 30 dBm with a PAE of more than 32%.

This paper is organized as follows. Section 1 introduces the challenges of broadband PA design and the concept of stacked FETs. Section 2 explains the design process based on the small-signal model to calculate the interstage matching network of stacked FETs, and the designed circuit is presented. Section 3 presents the measurement results and a comparison with other designs. A short conclusion is given in Section 4.

## 2. Circuit Design Considerations

In this paper, a PA MMIC covering the Ku-band was designed based on 0.15 µm GaAs HEMT technology. This technology is a 0.15 µm single-recess pHEMT process designed for high-frequency noise amplifier and medium-power amplifier products operating at up to 4 V. The technology demonstrates excellent device-level performance with an *F_t_* of 100 GHz. The process was designed with a maximum rating of a 4 V drain bias and exhibited typical breakdown voltages of 9 V, with a minimum of 8 V in process control monitoring. The process has two layers of metal in the interconnection line. The lower metal layer has a current tolerance of 4 A/mm, and the upper metal layer has a current tolerance of 6 A/mm. An air bridge connection pattern was used at the intersection of the two metal layers.

### 2.1. Overall Circuit Analysis and Design

Figure 1 shows a simplified circuit schematic of the designed PA and the detailed parameters of some key devices. It contains a power drive stage, as well as a power synthesis stage. The high-power output can be satisfied while ensuring high-power gain.

A single stacked FET B can reach a maximum output power of 29 dBm after load pull. To achieve the output power characteristic of one Watt, two stacked FET B need to be connected in parallel. Compared to a single HEMT, the gain of a stacked FET is larger, and the gain equivalent to a cascade of three HEMT transistors can often be achieved by cascading two stacked FETs. In order to improve the stability of the circuit, RC networks are used at the gates of the driver-stage transistors and output-stage transistors to improve the overall stability of the proposed PA. In addition, *L_D1_* and *L_D2_* as part of the matching circuit need to consider the current carrying capacity of the line width (especially the DC bias line in the output stage, so the line width of *L_D2_* was 50 µm in this design).

### 2.2. Impedance Analysis of Stacked Transistors

The detailed schematic of the circuit of a double-stacked power amplifier is shown in Figure 2, where two transistors are stacked to combine the drain–source voltage swing of each transistor. As the input power increases, the drain voltage is likely to be higher than the gate–drain breakdown voltage of the transistors. Therefore, a grounded capacitor is usually cascaded at the gate of the common-gate transistor, which allows part of the RF voltage swing to appear at both ends of the capacitor [5,7,19]. The value of the gate capacitance (*C_g_*) is determined by the optimal load impedance of the lower transistors so that each transistor can produce the maximum voltage swing and the maximum output power. For design convenience, the calculation of the *C_g_* value is given below [21].
(1)$$Z_{in}=\left(1+\frac{C_{gs2}}{C_{g}}\right)\cdot \left(\frac{1}{g_{m2}}∥\frac{1}{j\omega C_{gs2}}\right)\approx \left(1+\frac{C_{gs2}}{C_{g}}\right)\cdot \frac{1}{g_{m2}}$$
(2)$$C_{g}=\frac{C_{gs2}}{g_{m2}R_{opt}-1}$$
where *C_gs_* is the gate–drain parasitic capacitance, *g_m2_* is the transconductance of the upper transistor [22], and *R_opt_* is the load impedance of the lower transistor. To provide the best load line impedance for each transistor and ensure that the drain–source voltage is equally distributed among the stacked devices, the impedance *Z_in_* should be adjusted to *R_opt_*. In fact, the effect brought by the gate–source parasitic capacitance *C_gs2_* of the upper transistor at lower frequencies cannot be simply ignored. Therefore, *Z_in_* is not a real impedance, and the imaginary part of *Z_in_* is very unfavorable for impedance matching in the broadband range [21].

A simple and effective way to achieve impedance matching between the upper transistor and the lower transistor is by introducing an inter-stage matching structure. As shown in Figure 2, the impedance matching of the stacked transistors can be improved while interconnecting the upper and lower transistors by introducing an interstage matching inductor *L_m_*. A theoretical study was conducted to demonstrate the role of *L_m_* more intuitively. With the introduction of the interstage matching inductor, *Z_opt1_* can be expressed as:(3)$$\begin{array}{l} Z_{opt1}=\left(1+\frac{C_{gs2}}{C_{g}}\right)\cdot \left(\frac{1}{g_{m2}}∥\frac{1}{j\omega C_{gs2}}\right)+j\omega L_{m}\\ =\overset{\mathrm{real}\;\mathrm{part}}{\overset{︷}{\left(\frac{g_{m2}}{g_{m2}{}^{2}+\omega^{2}C_{gs2}{}^{2}}+\frac{g_{m2}C_{g}C_{gs2}}{g_{m2}{}^{2}C_{g}{}^{2}+\omega^{2}C_{g}{}^{2}C_{gs2}{}^{2}}\right)}}+j\omega \overset{\mathrm{imaginary}\;\mathrm{part}}{\overset{︷}{\left(L_{m}-\frac{C_{g}C_{gs2}+C_{gs2}{}^{2}}{g_{m2}{}^{2}C_{g}+\omega^{2}C_{g}C_{gs2}{}^{2}}\right)}} \end{array}$$

From (3), it can be seen that the imaginary part introduced by *C_gs2_* cannot be neglected when the frequency is low. However, the introduction of *L_m_* can reduce the imaginary part of *Z_opt1_*, which can further reduce the effect of the parasitic capacitance *C_gs2_* on the high-frequency impedance and facilitate the bandwidth extension to high frequencies [23]. Compared to the conventional direct interconnection structure, the structure with an interstage matching inductor in this design can significantly improve the bandwidth characteristics. The interstage matching inductor in this design was achieved by introducing two symmetrical microstrip connecting lines, where the inductance of *L_m1_* was approximately 0.29 nH at 16 GHz, and the inductance of *L_m2_* was approximately 0.18 nH at 16 GHz.

Compared to PAs with parallel transistor structures, the optimal output impedance of the transistor stacked FET structure is relatively large, so output matching is easier to achieve [22]. In addition, since the output impedance of the stacked FET structure is closer to the impedance matching point of 50 ohms, the output matching network can be realized in a low-Q region, which is a great advantage for wideband matching. In this design, the load impedance was increased by as much as a factor of two compared to current combinational PAs of similar power levels. To visualize the variation interval of the optimal load impedance, we performed a load pull on a single HEMT tube and the stacked FET B, as shown in Figure 3. It can be seen that the load impedance intervals for the optimal efficiency and power of the stacked FET structure was much closer to the impedance matching point.

For the driver-stage stacked FET A, a potential instability often occurs due to its large gain. To ensure the unconditional stability of the whole PA, a parallel RC network consisting of bypass capacitors and resistors was used at the input of the common-source transistor to suppress excessive low-frequency gain [22]. On the other hand, the common-source transistor is the main source of in-band instability, and the high-Q inductive gate feed line tends to cause the instability of the common-gate transistor at high frequencies (>30 GHz) [23]. For this reason, a small series resistance (*R_g1_* = 4.7 ohms) was inserted between the gate terminal and the ground capacitor to reduce the Q-factor of the inductive gate transfer line.

## 3. Measurement Results

The design was fabricated using a 0.15 µm GaAs pHEMT process. A micrograph of the chip is shown in Figure 4, with an area of 3.3 × 1.1 mm^2^ (including all input/output test pads). The test protocol included both small-signal testing and large-signal testing. On-chip small-signal testing of the S-parameters of the fabricated PA was performed using a probe stage (Cascade Summit 11,000 M) and a vector network analyzer (Agilent N5244A); large-signal testing was performed by eutectic soldering of the chip to a molybdenum-copper carrier to ensure good grounding and heat dissipation, followed by a signal source and power meter. The heat generation of this design was much lower than that of the PA in the GaN process, so the actual large-signal test was performed by surface-mounting the chip to the molybdenum-copper carrier, which was sufficient to support its heat dissipation needs and, therefore, did not require additional heat dissipation techniques. Under both test conditions, the PA was biased with VD = 8 V, VM = 3.3 V, and VG = −0.6 V.

Figure 5 and Figure 6 show the simulation and test results of the small-signal test. The test results showed that the input return loss was less than −5 dB and the small-signal gain was 31 dB at maximum. The actual tested S11 deteriorated significantly compared to the simulation results, and we summarize and analyzed some of the influencing factors as follows:In the simulation process of the PA, in order to ensure the consistency between simulation and actual application, we usually consider the influence of the gold wire bonding line in the simulation. However, the use of the probe table for small-signal testing did not introduce the gold wire bonding line, so there was a certain test error.Vector network analyzer calibration error (this affected the actual test results, but the impact was not significant, which was related to the actual operation of the tester).Model error: according to our past experience, the model provided by the manufacturer had a certain error, with the actual test results compared to the simulation effect.The impact of chip temperature (which tends to occur only in high-heat-generating devices, for example a PA): the increase in chip temperature often changed the electrical parameters of the chip, resulting in changes in the test results.

Figure 7 shows the simulation and measured results of the large-signal test. The measurements were performed under continuing-wave conditions. The measured results showed that, when the input power was 8 dBm, the saturation output power could reach 30 dBm in the range of 15–17.5 GHz; the saturation output power was greater than 27 dBm in the range of 14–19 GHz.

Figure 8 shows the simulation and measurement results of the PAE. The test results showed that the peak PAE of the PA was 42%.

The tested output power, PAE, and gain curves with the input power are given in Figure 9. According to the test results, it can be concluded that the output power gradually saturated as the input power increased. When the input power reached 2 dBm, the output power began to compress, and the PAE reached its maximum at an input power of 8 dBm.

Table 1 shows the comparison of our work with other PAs published in the literature. It can be found that the present design had a large advantage in terms of bandwidth and gain when compared with PAs of the same process. The present design is suitable for broadband radar systems.

## 4. Discussion and Conclusions

In this paper, a Ku-band broadband PA MMIC was designed based on 0.15 µm GaAs HEMT technology, which was fabricated using WIN’s semiconductor process. The advantages of stacked FET structure for broadband design were theoretically analyzed, and a PA with double-stacked transistors was designed. The experimental results were consistent with the design simulation results, which verified the feasibility of the proposed theory and design method. In the frequency range of 14–19 GHz, the proposed PA MMIC had a power gain of 22.8 dB, a maximum saturation output power of 30.8 dBm, and a peak PAE of 41%. In addition, the overall size of the proposed wideband PA MMIC was only 3.3 × 1.2 mm^2^, achieving excellent performance and meeting the requirements proposed for wideband radar systems.

## Figures and Tables

**Figure 1 micromachines-14-01276-f001:**
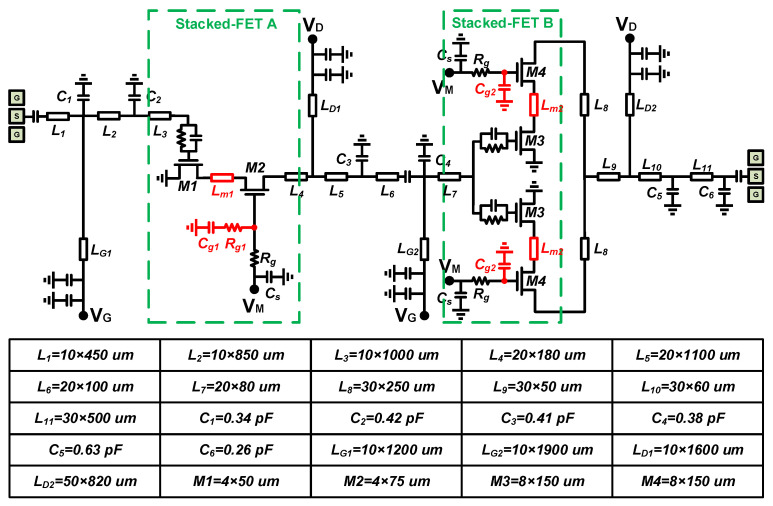
A simplified circuit schematic of the entire PA and the parameters of some key devices.

**Figure 2 micromachines-14-01276-f002:**
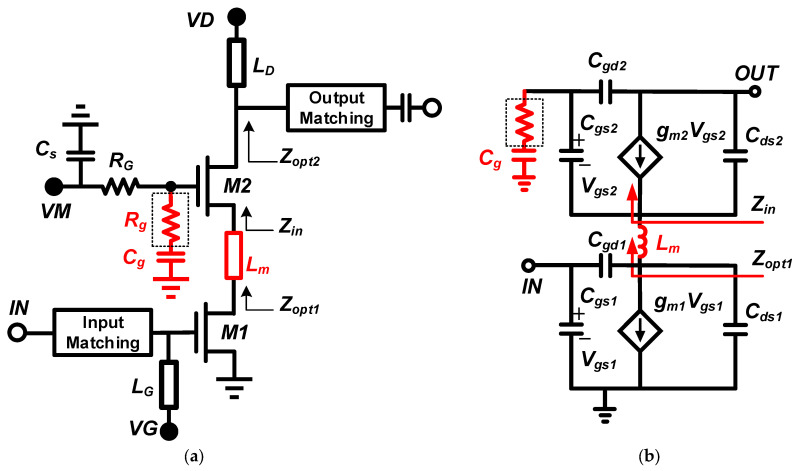
(**a**) Schematic of the double-stacked PA. (**b**) A simplified model of a single-ended double-stacked FET with parasitic capacitances.

**Figure 3 micromachines-14-01276-f003:**
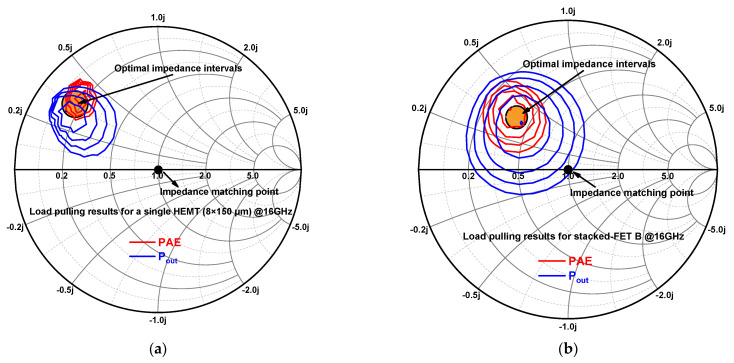
(**a**) Load pulling results for a single HEMT (8 × 150 μm). (**b**) Load pulling results for stacked FET B.

**Figure 4 micromachines-14-01276-f004:**
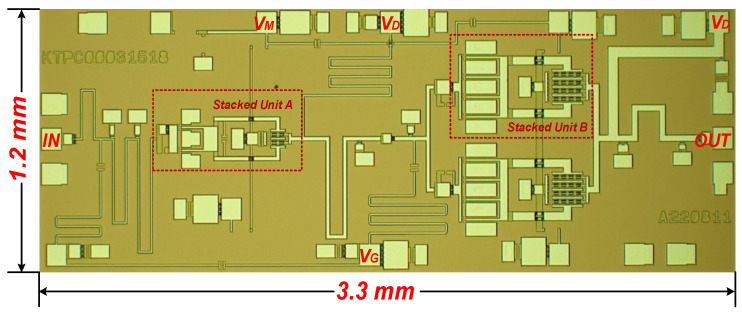
The micrograph of the proposed broadband PA MMIC.

**Figure 5 micromachines-14-01276-f005:**
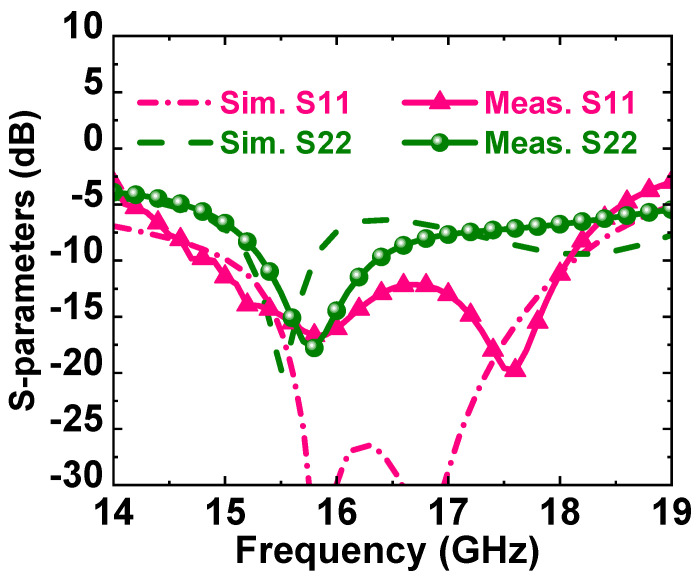
Simulated and measured S11 and S22 of the proposed broadband PA MMIC.

**Figure 6 micromachines-14-01276-f006:**
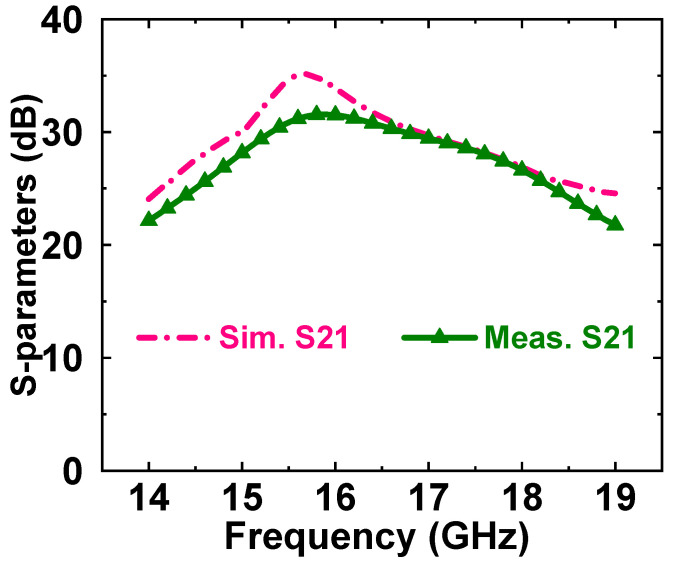
Simulated and measured S21 of the proposed broadband PA MMIC.

**Figure 7 micromachines-14-01276-f007:**
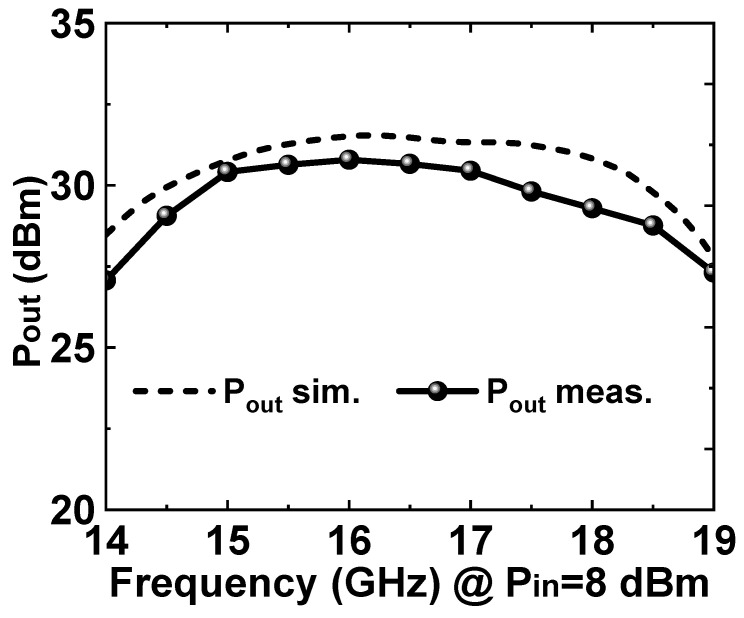
Simulated and measured saturated output power of the proposed broadband PA MMIC.

**Figure 8 micromachines-14-01276-f008:**
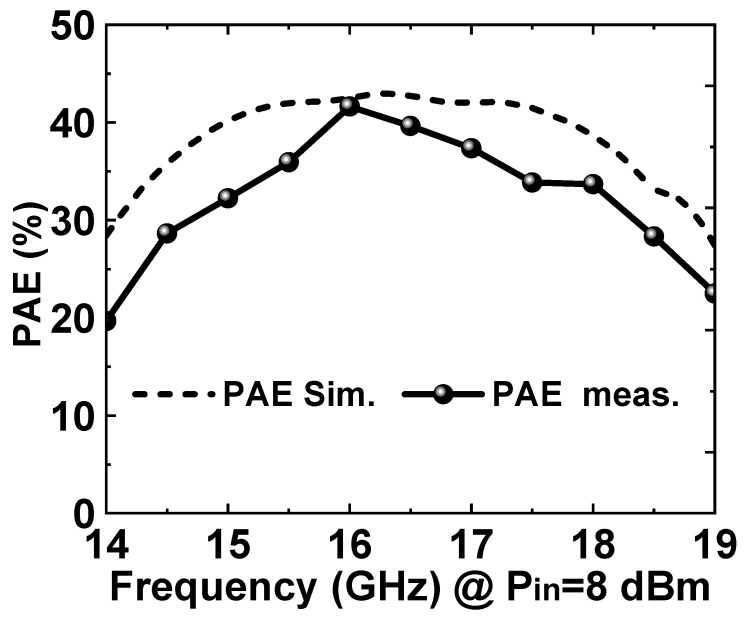
Simulated and measured PAE of the proposed broadband PA MMIC.

**Figure 9 micromachines-14-01276-f009:**
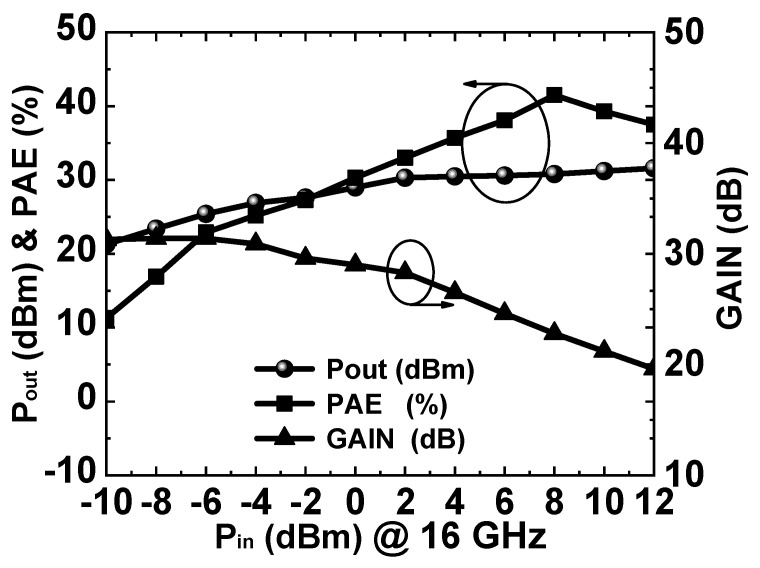
Measured Pout, PAE, and gain curves versus Pin at 16 GHz.

**Table 1 micromachines-14-01276-t001:** Performance of broadband multistage GaAs MMIC PAs.

Ref.	BW (GHz)	BW * (%)	Psat (dBm)	Gain ** (dB)	PAE
[22]	9–22	84	33.7	14	NA–29.5
[24]	10.5–16.5	44	35	18	36–41
[25]	26–31	18	31.5	16.7	21–33
[26]	20–23	14	31.8	9.5	NA–24
This work	14–19	30	30.8	22.8	20–41

* Defined as the ratio of the bandwidth to the center frequency. ** Power gain.

## Data Availability

Not applicable.

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
