# Peer review of "A Ku-Band Broadband Stacked FET Power Amplifier Using 0.15 μm GaAs pHEMT"

_micromachines, 2023, doi:10.3390/mi14061276_

Round 1

Reviewer 1 Report

The authors present a broadband power amplifier based on GaAs stacked pHEMT transistors for the Ku-band systems. The performance results are impressive and have been supported by the theoretical modeling. The structure and presentation of the paper is well organized and logically sound. However, the paper needs some modifications before it can be published in Micromachines Journal. The following issues have to be addressed:

1. The statements on lines 50-63 in page 2 of the manuscript needs a reference citation.

2. How was the inter-stage matching inductor implemented on the chip? The authors should mention that in the paper.

3. The S11 amplitude of the measured device much lower than that of the simulated S11. What reason could be attributed to it? It should be mentioned in that section.

4. It is not clear in the paper whether the chip implementation of the designed PA circuit system was fabricated by the authors or was the circuit design sent to a third party to be fabricated for them. If it is fabricated by the authors, there should be a separate section on materials and methods employed to fabricate the GaAs stacked pHEMPT transistors as well as a cross-sectional diagram of the stacked transistor device structure. 

After the above issues have been resolved, the paper can be considered for publication in Micromachines.

Reviewer 2 Report

Dear Authors.

I have only two remarks regarding your work:

1. Since you are presenting broadband amplifier, highlighting PAE of 41% achieved in a quite narrowband region is a bit misleading.
I would suggest to update table 1 with PAE values of your as well as other PAs  in a broader band - at least 20% Fractional Bandwidth. Also, narrowband PAE shouldn't be highlighted in the abstract as well.

2. Can you add temperature analyses?

In other words, how much is dissipation of your amplifier?

Does it require active cooling, etc.

Best regards,

Dejan

English language is fine.
